# The Second Derivative of Fullerene C_60_ (SD-C_60_) and Biomolecular Machinery of Hydrogen Bonds: Water-Based Nanomedicine

**DOI:** 10.3390/mi14122152

**Published:** 2023-11-25

**Authors:** Lidija R. Matija, Ivana Mladen Stankovic, Milica Puric, Milica Miličić, Danijela Maksimović-Ivanić, Sanja Mijatovic, Tamara Krajnović, Vuk Gordic, Djuro Lj. Koruga

**Affiliations:** 1Nano Lab, Department of Biomedical Engineering, Faculty of Mechanical Engineering, University of Belgrade, 11220 Belgrade, Serbia; imileusnic@mas.bg.ac.rs (I.M.S.); milica.milicic@tftnanocenter.rs (M.M.); 2TFT Nano Center, 11050 Belgrade, Serbia; 3Institute for Biological Research Siniša Stanković—National Institute of the Republic of Serbia, University of Belgrade, 11000 Belgrade, Serbia; nelamax@ibiss.bg.ac.rs (D.M.-I.); sanjamama@ibiss.bg.ac.rs (S.M.); tamara.krajnovic@ibiss.bg.ac.rs (T.K.); vukgordic6@gmail.com (V.G.); 4NanoWorld, Biomedical Photonic Lab, 11043 Belgrade, Serbia

**Keywords:** water, fullerene C_60_, icosahedral soft-matter, hydrogen bonds, Fibonacci water chains, nano/micro molecular machinery, cancer, nanomedicine

## Abstract

The human body contains 60–70% water, depending on age. As a body fluid, it is not only a medium in which physical and chemical processes take place, but it is also one of the active mediators. Water is the richest substance with non-covalent hydrogen bonds. Water molecules, by themselves (in vacuum), are diamagnetic but when organized into clusters, they become diamagnetic or paramagnetic. Also, biomolecules (DNA, collagen, clathrin, and other proteins) have non-covalent hydrogen bonds in their structure. The interaction, as well as signal transmission, between water and biomolecules is achieved through the vibrations of covalent and non-covalent hydrogen bonds, which determine the state and dynamics of conformational changes in biomolecules. Disruptive conformational changes in biomolecules, cells, and tissues lead to their dysfunctionality, so they are a frequent cause of many disorders and diseases. For example, the rearrangement of hydrogen bonding due to mitochondrial disease mutation in cytochrome bc1 disturbs heme bH redox potential and spin state. In order to prevent and repair the dysfunctional conformational changes, a liquid substance was developed based on the second derivative of the C_60_ molecule (SD-C_60_), which has classical and quantum properties. The characterization of SD-C_60_ by UV-VIS-NIR, FTIR, TEM, and AFM/MFM was performed and it is shown that SD-C_60_ water layers generate vibrations with near-zero phase dispersion which are transmitted through Fibonacci’s water chains to biomolecules. In comparison with previously published SD-C_60_ derivate (3HFWC, size until 10 nm, and 1–5 water layers), the improved formulation (3HFWC-W, size 10–25 nm, and 6–9 water layers) showed multiplied cytotoxic activity against melanoma cell lines of different aggressiveness. Apart from this, the mode of action was preserved and based on an induction of senescence rather than cell death. Importantly, high selectivity towards malignant phenotypes was detected. Observed effects can be ascribed to a machinery of hydrogen bonds, which are generated in SD-C_60_ and transmitted through water to biomolecules. This approach may open a new field in science and healthcare—a “water-based nanomedicine”.

## 1. Introduction

Water, compared with other natural polymolecular structures, has more than 35 “anomalies”. The term “anomaly” (under quotation marks) is used because compared with, for instance, 100 structures exhibiting one type of behavior, water behaves differently. One such example is that all materials contract in cold and expand in heat. Water behaves differently; liquid water has a greater density than ice, thus ice floats. Water expands at lower temperatures, and contracts at higher temperatures. Without this “anomaly”, biological forms of life probably would not exist. Water effected a breakthrough and created biological life, aided by external factors, primarily light, heat, and gravitation. Water within a cell makes up about 70% of the cell’s contents; the remaining contents are other chemicals, mostly proteins (15%). The organism as a whole contains approximately 72% water, the rest being dry matter. Lungs, blood, and the brain are most abundant with water. With ageing, the water percentage decreases in every organ, mostly within skin (from 70% to 60%). The interaction between water and biomolecules assists in molecular recognition, which makes water an active rather than a passive factor for biomolecular function [1].

The water molecule is composed of two hydrogen atoms and one oxygen atom, connected by covalent hydrogen bonds. The strength of the covalent bond O-H is 460 kJ/mol (4.77 eV), while the strength of the noncovalent bond O…H ranges from 10 to 120 kJ/mol. (0.09–1.24 eV) depending on the pKw, temperature, and the organization of the water molecules: dimer, trimer—open chains, clusters, crystalline linear form (liquid crystals), self-similar quasicrystals or Penrose tiles [2,3,4]. The organization of water molecules depends primarily on the angle between the hydrogen atoms covalently bound to oxygen in the water molecule. It was shown that both bond lengths and bond angels are related to icosahedral symmetry element Φ ($\frac{1}{2}\left(1+\sqrt{5}\right)\;=\;1.618033989\ldots )$ [5]. Moreover, it was shown that the fine structure of matter, which was defined by Sommerfeld (*α* = 2*π e*^2^/*hc*) = 0.007297…, where *e* is the elementary charge, *h*—Planck’s constant and *c*—speed of light), can also be defined by Φ (α = Φ^2^/(360 − 2/Φ) = 0.007297…, accuracy to six decimal places). Since the fine structure constant is considered a fundamental physical constant (inverse α^−1^ = (360 − 2/Φ)/Φ^2^ = 137.0359…), which quantifies the strength of the electromagnetic interaction between elementary charged particles, it indicates that there is a connection between the distribution of charges of valence electrons in molecules and distance proportion between atoms in molecules defined by Φ [6]. 

Two water molecules (dimer) or more (trimer or longer water chains) engage in interactions via noncovalent hydrogen bonds. The basic reaction is 2H_2_O ↔ [H_3_O]^+^ + [OH]^+^, where K_w_ = [H_3_O]^+^ × [OH]^−^, and pK_w_ = −log10K_w_. The rate of water molecules’ interaction, primarily of noncovalent hydrogen bonds and of the reorganization of dipole moments, is within limits 1–100 ps at a nano scale. The microscopic dynamics and the conformation changes in biomolecules occur within microseconds (μs), while on the macroscopic level, (large clusters) can be in range from milli seconds (ms) to a few seconds (s). It was demonstrated that charge distribution (donor/acceptor) and the length of the hydrogen bond in water and biomolecules are determined by Φ [7,8]. This new evidence is in line with Felix Franks’ claim (1972) [7,8] that “…of all known liquids, water is probably the most studied and least understood”.

The organization of water molecules in chains, clusters, and multi-shells possesses an element of icosahedral symmetry ($\frac{1}{2}(1+\sqrt{5\;})$ = Φ = 1.61803…, with particular solutions of Φ, as a set of Fibonacci sequences: 3/2 =1.5, 5/3 = 1.66, 8/5 = 1.60, 13/8 = 1.625, 21/13 = 1.615, 34/21 = 1.619, 55/34 = 1.617, 89/55 = 1.618, 144/89 = 1.617…… Appendix A). There is theoretical and experimental evidence that cold water may be organized as Penrose aperiodic tiles [9]. Water organization in quasicrystal is possible because the angle between hydrogen atoms in molecules can be from 104°30′ (liquid) to 109°17′ (ice) and may have a possible particular solution of Φ. However, Penrose ideal tiles have anglea of 108° (Figure 1a) and this is one of the reasons for icosahedral symmetry formation. On other hand, de Boissieu [9] said that Penrose tiling is not completely reliable in representing quasicrystal in three-dimensional space. In spite of this observation, de Boissieu wrote that three-dimensional Penrose tiling (3DPT) can be generated by the six-dimensional cut method ($E^{6}$ → $E_{1}^{3}+E_{2}^{3},$ where $E_{1}^{3}⊥E_{2}^{3}$, rhombohedra with four edges perpendicular to a fixed face of the dodecahedron) and that such “parallel” components have been experimentally observed for the first time [9]. However, icosahedral aperiodic tiling are well described as solid quasicrystal (alloys of three or more elements: Al, Mn, Fe, and Cr) by Dan Shechtman, who received the Nobel prize for chemistry in 2011 [10,11,12]. Based on theoretical and experimental results, it is shown that soft-matter quasicrystals with icosahedral symmetry may exist in biological systems [13]. There are huge differences in mechanical properties between solid quasicrystals and soft-matter quasicrystals. “For example, under action of impact tension with stress amplitude σ_0_ = 5 MPa, the variation of mass density δ*ρ*/*ρ*_0_ is 10^−14^ for solid quasicrystal, while under action of impact tension with stress amplitude σ_0_ = 0.01 MPa, the variation of mass density δ*ρ*/*ρ*_0_ is 10^−3^ for soft-matter quasicrystals” [13]. In this paper, we experimentally demonstrated using TEM and AFM/MFM that a very stable soft quasi-crystalline water shells (dry state) around C_60_(OH)_36_, in a form of icosahedral 3-dimensional structure, exists. Water layers composed of elastic soft-matter quasicrystal (deformable, similar to peptide planes of proteins) can be obtained, using sets of nine or sixteen different Fibonacci numbers from its sequence, which determines the water icosahedral structure (different angles between hydrogen atoms of water molecule according to 3 × 3 and 4 × 4 Fibonacci determinants, Appendix A).

In order to stabilize the water layers’ icosahedral structure of a soft substance, water molecules have to be organized around a hard quasi-crystal that will generate icosahedral vibrations that will be resonantly transmitted to the water layers. This will amplify the vibration amplitudes depending on the number of layers and transmit them to the surrounding water and to the biomolecules. Of all the existing molecular crystals with icosahedral symmetry, the C_60_ molecule turned out to be the best candidate as a vibrational nano generator of icosahedral symmetry.

From a physics point of view, the C_60_ is a spherical molecular crystal composed of 60 carbon atoms arranged in 12 pentagons and 20 hexagons. The C_60_ possesses high symmetry with 120 symmetrical transformations (higher than diamond which has 48), having high speed rotation and “twisting” movement (a billion per second), and it possesses wave-particle duality (quantum/classical properties) [14]. It also possesses self-harmonized attractive–repulsion forces through the vibration of carbon atoms of pentagons (diamagnetic) and hexagons (paramagnetic). 

From a chemistry point of view, the C_60_ molecule has both advantages and disadvantages. The advantages are the existence of icosahedral symmetry (as some biomolecules and complex biological structures), the ability to be functionalized, as well as to form small or large molecules, that it can be a carrier of drugs, and it can be used as a marker in diagnostics. The disadvantages are that it is not soluble in water, it is reactive (double bonds of hexagons can be open), and can be toxic, which depends on its concentration. 

In order to overcome some of the above mentioned disadvantages, the C_60_ molecule is funcionalized with OH groups (the first C_60_ derivative, FD-C_60_). Fullerene hydroxylation (FD-C_60_) increases water solubility and affects these nano particles’ interactions with biological systems. It is demonstrated that increasing fullerene water solubility through C_60_ surface modification leads to significantly decreased toxicity [15]. Specifically, in this study, the decreased toxicity of FD-C_60_ compared with cytotoxic effect of fullerene aggregates to human skin (HDP) and liver carcinoma (HepG2) cells was observed [15]. Similarly, it was observed that hydroxylation decreases the toxic potential of fullerene on mouse L929 fibrosarcoma, rat C6 glioma, and U251 human glioma cell lines [16]. Additionally, FD-C_60_ induced apoptotic changes on investigated cells lines, while fullerene C_60_ induced necrotic cell death [16]. The distinct effects of pristine and modified fullerene originate from the different nanoparticles’ interaction with the intracellular metabolic pathways [16,17].

However, FD-C_60_ did not provide the satisfactory absence of toxicity in living systems [15]. In order to reduce toxicity and even completely eliminate it, as well as to improve the transfer of vibrational signals of C_60_ molecules to biological water and biomolecules, the second derivative of the C_60_ molecule, SD-C_60_ (commercial name: 3HFWC—hyperharmonized hydroxylated fullerene water complex) was designed and made by creating water layers (shells) around FD-C_60_ (fullerol). The size of SD-C_60_ (3HFWC) varies from 3 to 30 nm, depending on the number of water layers [18,19].

Bearing in mind that, biochemically, FD-C_60_ showed beneficial effects in the treatment of cancer, the design and synthesis of SD-C_60_ was achieved, which reduced or even completely eliminated the toxicity of the original molecule (FD-C_60_). Since interactions with biomolecules are based on a biophysical approach, this will produce different effects to a biochemical approach.

Our goal is to develop a method that will reprogram malignant cells, rather than causing tumor cell death which might have serious consequences like triggering compensatory proliferation and tumor repopulation upon applied treatment in advanced tumor stages.

## 2. Materials and Methods

### 2.1. Water

Tap water, from Belgrade’s city water supply system, was used (Ca^2+^—71.1 mg/L, Mg^2+^—14.7 mg/L, Na^+^—11.4 mg/L, K^+^—1.0 mg/L, Fe^2+/3+^—0.05 mg/L, NH_4_^+^—0.03 mg/L, Cl^−^—14.1 mg/L and NO_3_^−^—6.1 mg/L), at a conductivity 85 mS/cm. Then, water was treated using reverse osmose (Ca^2+^—0.60 mg/L, Mg^2+^—0.53 mg/L, Na^+^—0.32 mg/L, K^+^—0.04 mg/L, Fe^2+/3+^—<0.005 mg/L, NH_4_^+^—<0.03, Cl^−^—0.18 mg/L and NO_3_^−^—0.12 mg/L), at a conductivity in the tank of 0.05 mS/cm, and it was used for production SD-C_60_.

It is well known that under normal conditions, free water is organized into dimers, trimers, tetramers, and slightly more complex water structures with lifetimes from 50 fs to 100 ps. However, in a limited space at the nano and micro level, water can be organized into water chains according to the laws in the elements of icosahedral symmetry (Appendix A), because the values 3/2, 5/3, 8/5, 13/8 … respectively, are 1.500, 1.666, 1.600, and 1.625, which “oscillatory” converge to the value Φ = 1.618… (Appendix A), i.e., to the element of icosahedral symmetry $\frac{1}{2}(1+\sqrt{5})$ with eigenvalues T_1g_ and T_2g_ (Appendix A).

### 2.2. Manufacturing of SD-C_60_

The FD-C_60_ (fullerol, purity 99.5%) was ordered from Solaris, Edmonton, AB, Canada, while SD-C_60_ was synthesized at two laboratories the NanoWorld, Belgrade and the TFT Nano Center, Belgrade. Before mixing with water, fullerol was pre-treated with UVC sterilizer. Ultra-pure water was obtained by purifying water from a water supply system through the reverse osmosis process. Certain amounts of 0.150 g fullerol was mixed with 1 L ultra-pure water for initial experiment (NanoWorld lab, Belgrade, Serbia) and 3 g fullerol (TFT NanoCenter, Belgrade, Serbia) was mixed with 20 L of ultra-pure water for commercial use. This mixture passed through the process of steering and was heated at 38 °C for 40 min and then pumped into magnetic vessels by filtration through the membrane filter of 220 nm. Under external action of +250/−92 mT oscillatory magnetic field, according to eigenvalues of icosahedral symmetry T1g, T1u, T2g, and T2u, which for symmetry elements C_5_, $C_{5}^{2}$, S_10_, and $S_{10}^{3}$ gave solutions ±½(1 + $\sqrt{5\;}$) and ±½(1 − $\sqrt{5\;}$) (Appendix A). Under the internal action of the vibrations of the C_60_ molecules and the vibrations induced by the external magnetic field according to the same law (principle: “between a rock and a hard place”) at 37 °C in the reactor, water layers (shells) were formed around fullerol. Depending on the production conditions, two basic types of SD-C_60_ can be synthesized: a light-yellow solution (with a small presence of free fullerol in the solution) and white (without free fullerol in solution), with different numbers of water layers.

### 2.3. UV-VIS-NIR

Water, FD-C_60_ (fullerol) and SD-C_60_ (3HFWC, both yellow and white), is characterized using a Lambda 500 spectrometer, Perkin-Elmer, Waltham, MA, USA (the NanoLab, Faculty of Mechanical Engineering, University of Belgrade) in the range of 250–3000 nm.

### 2.4. FTIR

Water, FD-C_60_ (fullerol) and SD-C_60_ (3HFWC, both yellow and white), is characterized using the Spectrum Spotlight 400 FTIR Imaging System, Perkin-Elmer, USA (the NanoLab, Faculty of Mechanical Engineering, University of Belgrade) in the range of 2500–14,000 nm.

### 2.5. TEM

The samples in liquid state were applied to the copper mash coated with carbon. After draying the samples, we analyzed and recorded on the CM12 Philips/FEI Transmission Electron Microscopy, Eindhoven, The Netherlands, magnification ×45,000 and ×60,000, installed at the Faculty of Biology, University of Belgrade (the sample was prepared 6 months ago) and on TEM, JEM 1400, JEOL, Tokyo, Japan, magnification ×120,000 up to ×200,000, installed at the Faculty of Agriculture, University of Belgrade (one prepared 8 months ago, the other 3 years ago).

### 2.6. AFM/MFM

The materials in liquid state, FD-C_60_ (fullerol) and SD-C_60_ (3HFWC), each in bottles of 100 mL, were prepared in TFT Nano Center, Belgrade. The concentration of fullerol in the first derivative of C_60_ was 0.15 mg/mL, and in 3HFWC, it was also 0.15 mg/mL of fullerol (as a precursor), around which water layers were formed. The color of fullerol in solution was light brown, and 3HFWC samples in solution were very light-yellow to white.

Solutions of 2 mL of each were applied to diamagnetic foils in each of the 5 samples. First, the sample was dried with SD-C_60_ (3HFWC) with a gradual increase in temperature, from 25 to 110 °C over the next two hours. During the drying time, a visual control was performed every 15 min and the temperature was measured. After two hours, samples with dry and SD-C_60_ (3HFWC) was removed and samples with fullerol were placed to dry. The procedure was repeated as in the previous case.

After drying, the samples with SD-C_60_ (3HFWC) had two characteristic areas: one was white–grayish (transparent scrum) and the other was medium brown, while the fullerol samples were dark brown. After drying, the samples were stored in dark closed containers and kept at room temperature until use.

Characterization of dry samples was performed using JSPM-5200, Scanning Probe Microscopy, JEOL, Japan, (installed at the NanoLab, Faculty of Mechanical Engineering, University of Belgrade). Two methods were used: (1) AFM (Atomic Force Microscopy), and (2) MFM (Magnetic Force Microscopy). Both techniques are non-invasive. AFM method is based on Van der Waal’s forces and London-type dispersive forces between tip and sample, while MFM, in the non-contact imaging mode, is based on magnetic dipole–dipole interaction between tip and sample (measuring ± deflection of tip “ϕ” in deg.). For the purpose of magnetic gradient investigation, specialized cantilevers, type HQ NSC18/Co-Cr Al BS (MikroMasch, Tallinn, Estonia), with force constants in the range between 1.2 and 5.5 N/m and with the resonant frequency range between 60 and 90 kHz, were used. Prior to experiments, the cantilevers were placed in the external magnetic field of 0.4 T in order to induce a magnetic field themselves. Scanning size of sample depended on the object size and its number, which were of interest. Bearing in mind that a fullerol is 1.2 nm in size (as single molecule) and could be up to 2–3 nm (as aggregates), the optimal scan size is between 30 nm and 100 nm. Expected size of SD-C_60_ (3HFWC), based on theory and initial experiment by TEM, is between 5 nm and 30 nm, so the optimal scan size should be between 100 nm and 300 nm.

### 2.7. Biological Study In Vitro

Materials: Culture media DMEM (Dulbecco′s Modified Eagle′s Medium) and RPMI-1640 (Roswell Park Memorial Institute-1640) supplemented with 20 mM HEPES, 2 nM L-glutamine, and 0.01% sodium pyruvate were bought from Capricorn Scientific GmbH (Ebsdorfergrund, Germany). Penicillin/Streptomycin solution was from Biological Industries (Cromwell, CT, USA). Fetal bovine serum (FBS), phosphate-buffered saline (PBS), dimethyl sulfoxide (DMSO), trypsin, crystal violet (CV), propidium iodide (PI), and carboxyfluorescein diacetate succinimidyl ester (CFSE), were obtained from Sigma-Aldrich (St. Louis, MO, USA). 3-(4,5-dimethythiazol-2-yl)-2,5-diphenyltetrazolium bromide (MTT) was from AppliChem (St. Louis, MO, USA). Paraformaldehyde (PFA) was purchased from Serva (Heidelberg, Germany). Annexin V-FITC (AnnV-FITC) was bought from BD Pharmingen (San Diego, CA, USA), Apostat was from R&D Systems (Minneapolis, MN, USA), and acridine orange (AO) was obtained from LaboModerne (Paris, France).

The tested substance SD-C_60_ (3HFWC-W) [19] was obtained from TFT Nano Center (Belgrade, Serbia). The initial concentration of the substance was 0.15 g/L, while the working solutions for in vitro treatment were made in a culture medium, immediately before use. According to the manufacturer’s instructions, the substance was stored at room temperature (RT), and protected from light due to photosensitivity, while the working solutions were made far from light sources.

This research was performed on two melanoma cell lines of murine origin and different level of aggressiveness—B16F1 and B16F10, and a mouse embryonic fibroblast cell line—NIH/3T3. Cell lines are commercially available and originate from the American Type Culture Collection (ATCC). Both melanoma cell lines were kept in HEPES-buffered RPMI-1640 medium supplemented with 10% heat-inactivated FBS, 2 mM L-glutamine, 0.01% sodium pyruvate, penicillin (100 units/mL), and streptomycin (100 μg/mL). Cells were grown at 37 °C in a humidified atmosphere with 5% CO_2_. Mouse embryonic fibroblast cells were kept in the same conditions in the DMEM medium.

### 2.8. Methods

#### 2.8.1. Viability Tests

For viability determination, B16F1, B16F10, and NIH/3T3 cells were seeded at 1 × 10^4^, 8 × 10^3^, and 1 × 10^4^ cells/well density in 96-well plates, respectively. After 24 h, all tested cells were incubated at 37 °C in the absence or presence of a wide range of concentrations (0.09–6 µg/mL (for B16F1 and B16F10) and 0.1–150 µg/mL (for NIH/3T3)) of experimental substance 3HFWC-W for 24 h. At the end of the incubation time, two different cell viability tests were performed. The MTT assay is a colorimetric test based on the ability of cells to reduce the tetrazolium salt to formazan, a purple precipitate whose amount correlates with the number of living cells [20]. Crystal violet assay is a colorimetric test based on the binding of the base dye crystal violet to all negatively charged polysaccharides, proteins, and nucleic acids, where the amount of bound dye correlates with the number of viable cells [21]. Namely, at the end of the treatment, the supernatants were discarded. For the MTT test, cells were incubated at 37 °C with MTT solution (0.5 mg/mL) for approximately half an hour until purple formazan crystals were formed. Afterwards, the produced formazan was dissolved in DMSO. For the CV test, after supernatant disposal, cells were fixed with 4% PFA for 10 min at RT and then stained with 0.02% CV solution for 15 min. After washing with tap water, the dye was dissolved in 33% acetic acid. The intensity of dissolved color in both tests was evaluated by measuring the absorbance with an automated microplate reader at 540/670 nm. The viability of treated cells is represented as a percentage of the absorbance value of the control culture grown in the medium, which was arbitrarily assigned a viability value of 100%. The IC_50_ values were calculated using a four-parameter logistic function and presented as the mean ± standard deviation (SD) of three independently performed experiments.

#### 2.8.2. Flow Cytometry

Flow cytometry (Fluorescence Activated Cell Sorting, FACS) is a fast and reliable method with high sensitivity for the detection of discrete subpopulations of cells in large samples, which is based on fluorescent labeling of cells and measurement of fluorescence intensity [22]. For flow cytometry analysis, B16F1 and B16F10 cells were seeded at 3 × 10^5^ and 2 × 10^5^ cells/well density in 6-well plates, respectively. After 24 h, cells were incubated at 37 °C in the absence or presence of the IC_50_ value of experimental substance 3HFWC-W for 24 h. The measurement of the fluorescence intensity of labeled cell samples was performed using a CyFlow^®^ Space flow cytometer (Partec GmbH, Münster, Germany) and a CytoFLEX^®^ flow cytometer (Beckman Coulter, Pasadena, CA, USA), and the analysis of the obtained results was performed using the FlowJo^TM^ software program (https://www.flowjo.com/, accessed on 17 November 2023). 

Ann V-FITC/PI staining

For the detection of type I programmed cell death, double staining of cells with fluorescently labeled recombinant annexin V (Ann V-FITC) and propidium iodide (PI) was performed, which enabled the detection of populations of early-apoptotic (AnnV^+^/PI^−^) and late-apoptotic/necrotic (AnnV^+^/PI^+^) cells [23]. After the treatment, cells labeled with Ann V-FITC and PI, according to the manufacturer’s instructions, were incubated for 15 min at RT in the dark. The staining reaction was stopped by adding Annexin V Binding Buffer.

Apostat staining

FITC-conjugated pan-caspase inhibitor (ApoStat), which can passively diffuse through the cell membrane and irreversibly bind to activated forms of these enzymes, was used to identify and quantify total caspase activity in cells [24]. At the end of the treatment time, the cells were labeled with Apostat according to the manufacturer’s instructions and incubated at 37 °C for 30 min.

Acridine orange staining

To detect the process of autophagy, acridine orange (AO) dye was used, which binds with high affinity to acidic vesicles-autophagosomes, the characteristic markers of autophagy [25]. Upon the completion of the treatment, cells were stained with a solution of 10 μM AO for 15 min at 37 °C.

CFSE staining

Carboxyfluorescein diacetate N-succinimidyl ester (CFSE) was used to determine the influence of the tested substance on the proliferative potential of cells. This dye diffuses passively into the cell and binds covalently to the intracellular molecules. With each division, the fluorescence intensity in the daughter cells becomes twice as weak, which is suitable for monitoring the rate of cell proliferation [26]. Melanoma cells were stained with 1 µM CFSE for 10 min at 37 °C before the seeding and the treatment with 3HFWC-W. At the end of the treatment period, the intensity of green fluorescence was measured.

DHR 123 staining

The production level of reactive oxygen species (ROS) and reactive nitrogen species (RNS) was determined using the redox-sensitive dye dihydrorhodamine 123 (DHR 123), a non-fluorescent dye that freely diffuses into the cell and in the presence of ROS/RNS, oxidizes to fluorescent rhodamine 123. In this way, the total production of ROS/RNS (predominantly peroxynitrite and hydrogen peroxide) in the cells during the treatment is detected [27]. Melanoma cells were incubated for 20 min at 37 °C in a 1 µM solution of DHR 123 dye before the seeding and the treatment with 3HFWC-W. Following the end of the treatment period, the intensity of green fluorescence was measured.

DHE staining

Dihydroethidium (DHE) emits blue fluorescence in the cytosol where it can be oxidized in the presence of superoxide anion into ethidium, which then intercalates in DNA, emitting red fluorescence. By measuring the intensity of fluorescence on a flow cytofluorimeter, it is possible to determine the level of superoxide anion production in cells [27]. At the end of the treatment period, cells were stained with 20 µM DHE for 30–45 min at RT, in the dark.

#### 2.8.3. Microscopy

To detect and identify morphological and intracellular changes at the microscopic level, B16F1 and B16F10 cells were seeded at 5 × 10^4^ and 3.5 × 10^4^ cells/well density in 8-well chamber slides, respectively. After 24 h, cells were incubated at 37 °C in the absence or presence of the IC_50_ value of experimental substance 3HFWC-W for 24 h.

Light microscopy—staining with Oil Red O dye

For cell morphology analysis at the level of light microscopy, seeded and treated cells were observed without prior staining and digitally photographed on a ZOE Fluorescent Cell Imager (Bio-Rad Laboratories, Hercules, CA, USA). Alternatively, cells were stained with Oil Red O, a diazo dye used for staining and detection of lipid droplets, as well as lipofuscin granules. The staining was performed according to the manufacturer’s instructions using a commercially available kit (Bio Optica, Milano, Italy). After the 24 h treatment, the cells were washed with PBS and fixed in 4% PFA for 15 min at RT. The cells were then washed with distilled water and stained with Oil Red O solution for 20 min at RT. Contrast staining was performed with hematoxylin for 30 s, after which the plate was washed with running water for 3 min. Cells were mounted in a water-based mounting medium (BioMont Aqua, Biognost, Zagreb, Croatia). The microscopic preparation was observed on a Zeiss AxioObserver Z1 inverted fluorescence microscope (Carl Zeiss AG, Oberkochen, Germany) and photographed at 200× magnification.

Fluorescence microscopy—propidium iodide staining

Cell labeling with propidium iodide dye was used for the detection of morphological features of apoptosis at the level of fluorescence microscopy. Propidium iodide is a fluorescent intercalating dye that binds to the DNA molecule in cells, allowing visualization of the shape of the cell nuclei using fluorescence microscopy [28]. After the 24 h treatment, the cells were washed with PBS and fixed in 4% PFA for 15 min at RT. Staining was performed with PI solution (0.1% Triton X-100, 0.5 M EDTA pH 8, 50 µg/mL RNase, and 50 µg/mL PI—final concentrations in PBS) for 1–2 min at RT. After washing in PBS, cells were mounted in a fluorescence microscopy medium (Fluoromount-G™, eBioscience, San Diego, CA, USA) and analyzed on a ZeissAxio Observer Z1 inverted fluorescence microscope (Carl Zeiss AG, Oberkochen, Germany) at 200× magnification.

#### 2.8.4. Statistical Analysis

The data are presented as the means ± SD of at least three independently performed experiments. To calculate the statistical significance of the difference between the obtained results, an analysis of variance (ANOVA) was used, followed by the Student *t*-test for multiple comparisons. A value of *p* < 0.05 was considered statistically significant.

## 3. Results and Discussion

### 3.1. Water Layers

Water molecules can be organized in several forms: chain, graph, or closed structure (clusters and shells). In this paper, an organization in the form of icosahedral symmetry is considerd (according to the lengths of the hydrogen bonds and the angles between the covalent bonds of the hydrogen atoms in the water molecule) (Penrose tiling), which is deformable around the plane of symmetry (diagonal) and with the structural-energy equilibrium point A (Figure 1). In order to obtain a stable 3-dimensional structure from several of these deformable 2-dimensional icosahedral structures, it is necessary that, during these deformations, the change of angles is consistent with Φ, which is the structural-energy constant of icosahedral structures defined in this way (Penrose icosahedral tiling). Since the angle between two hydrogen atoms in a water molecule can be between 104.5 degrees (liquid) and 109.28 degrees (ice), the angle change should be in accordance with the set of particular solutions from the Fibonacci sequence Φ (for example, the Fibonacci sequence 3, 5, 8, 13, 21, 34, 55, 89, 144) whose determinant is equal to zero, giving the angle change from 105°14′ to 108°33′, with the fact that angles 104°30′ and 109°17′ will not be taken as they belong to the borderline cases of liquid and solid state of water (Figure 2). The packing system of deformable icosahedral water structures have to be orthogonal, because the pairing (joining) of the plates determines the oscillatory process of the magnetic field, that is, the state energy of covalent and non-covalent hydrogen bonds. The attractor of this effect is Φ/ϕ spirals (left and right orientation), whose discrete values are arranged like berries (seeds) in sunflowers (Figure 3, right). 

“Water tiles” (Φ-rhombs and pentagrams) are deformable 2D structures in a 3D space, similar to peptide planes in proteins. They are the realization of 3D Penrose tiling (3DPT) [9], not only for the angle 108° (Figure 1) but for all possible states according to a set of Fibonacci sequences of Φ in the interval 105.5 degree and 108.28 degrees (Figure 2).

One of the important elements of this process is the dynamics of charge redistribution between oxygen and hydrogen atoms, both in covalent and non-covalent bonds (Figure 4). If and only if (IFF) water molecules are in the process of forming an icosahedral water structure, under the influence of the oscillatory field (electric or magnetic) Φ and ϕ, can a harmonized charge distribution be established in the water system, thus the water layers will be formed. Experimental results based on neutron diffraction showed that, under certain conditions in biological systems, hydrogen bonds (covalent and non-covalent) can be coordinated by Φ (Appendix A) [29,30]. Also, theoretical research based on the analysis of experimental data points to this fact [5,6].

The precursor for obtaining SD-C_60_ (3HFW) is FD-C_60_ (fullerol), which is composed of the C_60_ molecules and a certain number of OH groups (in our case, the average number is 36, although with minor variations, it can be between 24 and 48). FD-C_60_ is a powder, yellow in color, and is well soluble in water. Under the influence of the C_60_ vibrations and the external oscillatory magnetic field (principle “between a hammer and an anvil”, Appendix A), water shells (soft-matter, layers) are formed around the FD-C_60_ (Figure 5).

During the production of SD-C_60_ (as a solution), under the influence of an oscillatory magnetic field, different forms of organization may be created (Figure 6). Fullerol in solution is present in about 0–0.05%. In spite of this, we tried to transform all fullerols to SD-C_60_ with water layers, but a small percentage could not be transformed. This is the reason why sometimes SD-C_60_ (3HFWC) as a solution has a light-yellow color. Fullerol with water layers (SD-C_60_ as soft-matter, quasicrystal) makes up about 2.5–3.0% of the solution: it has a compact elastic molecular structure with six water layers on average (from one to nine). When all fulleroles have water layers (and become SD-C_60_), then the solution turns white. In an SD-C_60_ solution, there is free water with about 37% (it is dimers, trimers, and higher water molecular organization). The composition of SD-C_60_ (3HFWC) is as follows: fullerol (0.05%) + fullerol with water layers (2.95%) + ordered water in Fibonacci chains (60%) + free water (37%).

If more than 0.05% FD-C_60_ (fullerol) remains in the SD-C_60_ solution, then the second derivative is SD-C_60_-Y (commercial name 3HFWC) with a light-yellowish color. If there is less than 0.05% of the first derivative (FD-C_60_) in the solution, then the SD-C_60_-W is white (commercial name 3HFWC-W).

In order to minimize the influence of minerals on the formation of an adequate network of water molecules in the water layers in the system, high purity water (0.055 µS/cm) was used. The temperature at which water layers are formed is around 37 °C. The pressure in the system is atmospheric, and the volume is 1 L (for experiments) and 20 L (commercial use), respectively. Applied external magnetic field is oscillatory (+250/−92 mT) according to eigenvalues of icosahedral symmetry T1g, T1u, T2g, and T2u, which for symmetry elements C_5_, $C_{5}^{2}$, S_10_, and $S_{10}^{3}$ provide solutions ±½(1 + $\sqrt{5\;}$) and ±½(1 − $\sqrt{5\;}$) (Appendix A).

When water molecules find themselves in the vibrational field of C_60_ molecules and in an external oscillatory magnetic field with the specified icosahedral symmetry elements (principle “between a hammer and an anvil”, Appendix A), then the water molecules, which possess dipole moments, will arrange themselves to icosahedral form, and after to a complex icosahedral soft-matter structure that has its own energy states of icosahedral symmetry (T_1g_, T_1u_, T_2g_, and T_2u_). As there are four different values (Φ, −Φ, ϕ, −ϕ, Appendix A), water molecules will be organized into four basic structures that are compatible between themselves, according to symmetry law.

Primer water cell (Φ-rhomb) is composed of four water molecules; two of them having hydrogen covalent bonds and in the other two, water molecules of adjacent water cells (non-covalent hydrogen bonds) will be organized depending on the angles between hydrogen atoms (which can be between 104.5 degrees and 109.28 degrees). This also means that the ratio of the length of the O-H and H…O bonds is flexible (from 96 pm to 106 pm and from 170 pm to 180 pm, respectively) [5,6]. The connection of the first layer with fullerol is through OH groups, of which there are about 36 on average.

### 3.2. Characterization of SD-C_60_ by UV-Vis-NIR and FTIR

In order to investigate hydrogen bonds in SD-C_60_, UV-ViS-NIR and FTIR spectroscopies were performed (Figure 7 and Figure 8). From the given diagrams, it can be observed that SD-C_60_ has a very pronounced peaks in the 2700–2750 nm range, which represents hydrogen bonds. If the peaks of SD-C_60_ are compared with the peaks of free water, one can see that the difference in their intensity is up to three times. This difference in intensity is the consequence of the water layers that are formed, which make SD-C_60_ a unique substance that can modulate the intensity of hydrogen bonds and, thus, affect the surrounding water (free water) and biomolecules that have hydrogen bonds in their structure.

In order to inspect and confirm observed difference between SD-C_60_-Y (with a small percentage of fullerol, FD-C_60_, in the free state in the solution) and SD-C_60_-W (without fullerol in the free state of the solution), NIR spectroscopy was used in the domain of the second overtone of water because absorption intensities in the wavelength ranges 1360–1366 nm and 1380–1388 nm are indicators of the existence or absence of water shells (layers) (Figure 9) [31,32]. As can be seen from the diagram (Figure 9, right), there are very pronounced intensities at the wavelengths 1360 nm and 1382 nm, which show that SD-C_60_-W has about twice as many water layers as the SD-C_60_-Y.

The diagram, Figure 10, shows the results of the C_60_, C_60_ (OH)_36_, and SD-C_60_ absorbance tests. As can be seen from the diagram, the hydrogen bond peaks of SD-C_60_ (domain around 2700 nm, no. 9) are several times more intense than those of fullerol (peak no. 3). It can also be observed that SD-C_60_ has another characteristic peak (no. 10) in the domain that corresponds to Amide-I vibrations. Two small peaks, no. 11 and no. 12, of SD-C_60_ indicate the presence of C_60_ (peaks no. 1 and no. 2).

### 3.3. Characterization of Dry SD-C_60_ by TEM

Since the SD-C_60_ solution contains soft matter, water arranged in Fibonacci chains, free water, and fullerol (in traces), it is necessary to dry the solution in order to obtain dry soft-matter. It was demonstrated, using TEM, that the size of the dry matter from a three-year-old solution (Fibonacci determinant D3 [18]), was in the range 3–10 nm, while the new substance (Fibonacci determinant D4 [19]) was in the range of 10–30 nm (Figure 11 and Figure 12).

### 3.4. Characterization of Dry SD-C_60_ by AFM/MFM

In order to identify water molecules as soft matter and precisely determine their size, the dry SD-C_60_ were examined using AFM and MFM (Figure 13 and Figure 14). The obtained results unequivocally prove that the soft matter of SD-C_60,_ in the dried state contains water molecules, because the dipole–dipole interactions (between the tip of the MFM probe and the dried soft matter) are many times greater than in the case when it is only present moisture in the dry matter (Figure 15 and Figure 16, and Appendix A).

### 3.5. Machinery of Hydrogen Bonds in Biomolecules

The existence of hydrogen bonds in biomolecules (Figure 17) is a well-known fact. However, the importance of hydrogen bonds for the functioning of biomolecules, cells, and tissues has not been given enough attention in the scientific community. Even though, in 1939, Linus Pauling pointed out the importance of hydrogen bonds with the words: “I believe that as the methods of structural chemistry are further applied to physiological problems, it will be found that the significance of *the hydrogen bonds* for physiology is greater than that of any other single structural feature”, hydrogen bonds did not attract enough heed. The machinery of hydrogen bonds in biological systems is a complex process and they primarily affect the change of conformational states and regulatory processes of biomolecules. Since the primary impact of SD-C_60_ on biological structures and processes is precisely through hydrogen bonds, biomedical experiments are needed. Bearing in mind that the oscillatory process of electric charge redistribution between oxygen atoms (or nitrogen) and hydrogen atoms is harmonized in hydrogen bonds (Figure 4), then if the violation of the oscillation of hydrogen bonds occur or their breaking in biomolecules, this biomolecule will become dysfunctional (Figure 18). Dysfunctionality of hydrogen bonds arises as a consequence of the symmetry-bracketing of their functionality (Appendix A).

### 3.6. Evaluation of Anti-Melanoma Effects of SD-C_60_-W (3HFWC-W) In Vitro

To investigate the effect of SD-C_60_-W (3HFWC-W) on the viability of mouse melanoma cells, two generations of B16 cell lines—B16F1 and B16F10, which are different in their invasive potential—were exposed to a wide range of concentrations (0.09–6 µg/mL) of tested substance for 24 h. At the end of the incubation period, cell viability was assessed using MTT and CV tests. As presented in Figure 19, 3HFWC-W led to a significant decrease in the number of viable cells in a dose-dependent manner, equally potent in low and highly aggressive melanoma cell lines (Figure 19A). In parallel, primary transformed cell line—NIH/3T3, representing normal healthy cells, showed significantly lower sensitivity to the same treatment, thus demonstrating the selectivity of the experimental item toward neoplastic cells (Figure 19A). This is well illustrated by IC_50_ values calculated from both viability screenings—MTT and CV in all tested cell lines (Table 1).

To evaluate the basic mechanism behind the observed cytotoxic effect, the presence of cell death, caspase activity, and cell proliferation were explored using Ann V-FITC/PI, Apostat, and CFSE staining, respectively (Figure 19B,D,E). In addition, the staining of cells with PI in chamber slides was performed in order to visualize the nuclear morphology of treated cells and compare it with an untreated control using fluorescence microscopy (Figure 19C).

Ann V-FITC/PI double staining revealed neither early (Ann^+^/PI^−^) nor late (Ann^+^/PI^+^) apoptotic cell accumulation in cultures exposed to the tested item, indicating that other mechanisms involved in 3HFWC-W-triggered cell viability decrease (Figure 19B). The PI staining of cells in chamber slides visualized the presence of large nuclei with a specific distribution of eu- and heterochromatin, and concordantly, confirmed the absence of apoptotic nuclear morphology in response to the 3HFWC-W. This effect was obvious in both cell lines with more prominent characteristics in an advanced form of melanoma—B16F10 (Figure 19C). Despite the absence of apoptotic cell death, the measurement of total caspase activity revealed the moderate activation of these molecules upon the treatment (Figure 19D). This effect is not rare since caspases are involved in heterogeneous processes, and there is no linear correlation between their activation and apoptosis as the outcome of the treatment [34]. In parallel, the assessment of the autophagic process did not report any difference between control and treated cells (Appendix A). Autophagy is the phenomenon of intracellular damaged structure self-digestion with dual role, varying from reparation of cell structures in order to save it from death, to taking a part in cell death realization [35]. The lack of differences in the dynamic of this process between control and treated cells indicated that tested nanoparticle influenced intracellular molecules in a very delicate manner. Further analysis of cell division rates using CFSE staining specified decreased capacity of B16F1 and B16F10 cells to proliferate when they are exposed to 3HFWC-W (Figure 19E). Behind all the mentioned effects, an enhanced intracellular production of hydroxyl radicals, peroxynitrite, as well as a superoxide anion, was detected using specific dyes-DHR 123 and DHE, respectively (Figure 20A,B). Keeping in mind oxidative species involvement in the regulation of intracellular/extracellular signaling at one side, and pathological conditions when it is not compensated by the cell redox protection [36] at the other, the observed potentiation of their production in the cells upon the exposure to 3HFWC-W suggested their arbitration in tested substance activities. Importantly, the absence of apoptosis as well as autophagy indicated that this experimental nanoparticle realized its activity in a sophisticated manner, apart from its high cytotoxic potential.

Large nuclei with spatial redistribution of heterochromatin resembling Senescence-Associated Heterochromatin Foci (SAHF) [37] in cells exposed to the treatment with an investigated item were in concordance with the change in cell size, shape, and overall morphology observed by light microscopy of vital cells (Figure 21, upper panel). All these specific features of the cell phenotype established in cultures upon only 24 h of incubation in the presence of 3HFWC-W, suggested that cells entered the state of accelerating aging known as cell senescence [38]. To confirm this, cells were stained with Oil Red O dye, a fat-soluble dye that stains neutral lipids/lipofuscin or “age pigment”, one of the strongest hallmarks of aging cells. Lipofuscins are products of oxidatively modified proteins and lipid degradation residues that accumulate in lysosomes [39]. As presented in Figure 21 (lower panel), a large quantity of lipofuscin granules marked by oil red color were detected in both tested cell lines, again with dominant accumulation observed in more aggressive B16F10 cells. Accordingly, it was evident that 3HFWC-W realized its anti-melanoma action basically through the initiation of cell reprogramming demonstrated by the senescence establishment, with a minor contribution of cell death.

Apart from the multiple improved cytotoxic effects, the main mechanism emphasizing the suppression of melanoma cell growth was the same, if we compare the effects of a new form of nanoparticle-SD-C_60_ (3HFWC-W) with a previously tested form—SD-C_60_-Y (3HFWC) [40,41]. Thus, following the obtained data, it can be expected that the upgraded formula of SD-C_60_-Y (3HFWC)–SD-C_60_-W (3HFWC-W) will demonstrate greater potential in comparison with the previously tested one, with the same physiological background which is already characterized as more desirable than conventional nonselective chemotherapy in advanced melanoma treatment.

## 4. Conclusions

A Penrose rhombic tiling, based on Φ (icosahedral symmetry), has two characteristic angles between hydrogen atoms in a water molecule; 108° and 72° (hydrogen covalently connected to oxygen and non-covalent connected, respectively). An angle of 108° is also an angle of pentagonal water order. In this paper, it was found that there are nine or sixteen types of stable particular solutions of Φ ({9,16}_Φ_) and that they synergistically fit the 3DPT (three-dimensional icosahedral Penrose tilings) much better than only one type of rhombic tiling with a 108° angle.

On the basis of 3DPT{9,16}_Φ_, the water icosahedral structure around the first derivative of C_60_ molecules (FD-C_60_) and thus the second derivative of the C_60_ (SD-C_60_), are explained. Water layers in SD-C_60_ were experimentally identified in solution (NIR and FTIR spectroscopy) and in a dry state in the form of soft-matter (TEM and AFM/MFM).

Two forms of SD-C_60_ with different numbers of water (shells) layers: from one to five (SD-C_60_-Y, commercial name 3HFWC) sizes up to 10 nm, and from six to nine (SD-C_60_-W, commercial name 3HFWC-W) with sizes 10–30 nm, were synthesized and characterized using UV-Vis-NIR spectroscopy. Their difference in color, number of water layers, and effects on biological structures were shown.

The existence of water layers, for the concentration of SD-C_60_ in the range from 1 to 10 mg/mL, eliminated toxicity, extended viability, and improved the effects on biological water and biomolecules (with hydrogen bonds; proteins, DNA, etc.).

The obtained results show that the main mechanism emphasizing the suppression of melanoma cell growth in vitro was the same for SD-C_60_-Y (3HFWC) and SD-C_60_-W. Upgraded formula of 3HFWC-3HFWC-W demonstrated greater potential to diminish cell growth in comparison with the previously tested one, while the induction of senescence might be considered a more desirable mechanism of antitumor action than conventional nonselective chemotherapy in advanced melanoma treatment.

Bearing in mind the beneficial effects of SD-C_60_ on murine cell lines, presented in this paper, as well as previously demonstrated effects on melanoma models in vitro and in vivo [40,41], it can be speculated that nano SD-C_60_ substances can open a new era for the treatment of different pathologies, called “water-based nanomedicine”.

## Figures and Tables

**Figure 1 micromachines-14-02152-f001:**
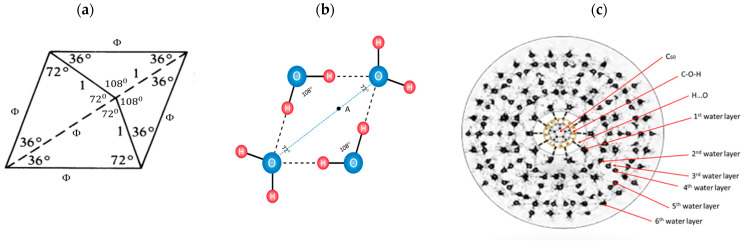
Schematic illustration of similarity of a Penrose tiling cell (base Φ—rhombi exhibiting icosahedral symmetry) (**a**), with one of several types of four water molecule ordered by Φ (**b**), as the generator of formation the complex icosahedral water multi-shells, (cross section, onions with 28, 60, 100, 280, 320, etc., water molecules) around FD-C_60_. (**c**). Cyclic water pentamers have bond angles of 108°, the same as the Penrose tiling cell. Thicker O-H lines are covalent and thinner dash O…H lines are non-covalent hydrogen bonds. The angle between two hydrogen atoms in a water molecule can be between 104.5 degrees and 109.28 degrees. Under the room conditions, 104.5 degrees is for liquid water and 109.28 degrees for ice (solid state). Under the influence of external factors, like temperature and oscillatory electric or magnetic fields, according to the law of dynamic ratio of covalent and non-covalent hydrogen bonds (Φ), the angle can be changed in the range of about 4°47′ (from 104°30′ to 109°17′). Rhombic tiling is deformable (elastic, soft-matter) around point A on the diagonal of the rhomb. Every water shell has 12 water pentagons and a different number of water hexagons (according to Euler polyhedral law).

**Figure 2 micromachines-14-02152-f002:**
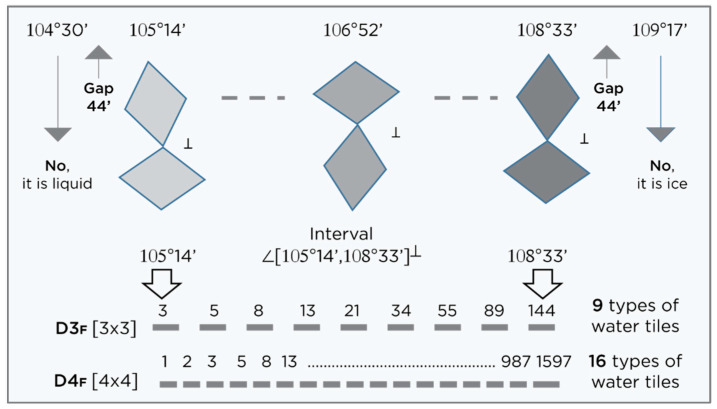
Bearing in mind that extreme states (liquid water and ice) are not desirable in the water layer around FD-C_60_, the states of water molecules with angles of hydrogen atoms 104°30′ and 109°17′ are excluded. The distance between these two extreme states and “water tiles” should be at least 0°44′, so that the soft quasicrystal state of water with icosahedron symmetry starts at 105°14′ and ends at 108°33′ values (⊥—system is perpendicular, D3_F_ = det
$\left|F3\right|$, determinant 3 × 3 of the Fibonacci sequence 3, 5, 8, 13, 21, 34, 55, 89, 144 and D4_F_ = det
$\left|F4\right|$, determinant 4 × 4 of the Fibonacci sequence 1, 2, 3, 5, 8, 13, 21, 34, 55, 89, 144, 233, 377, 610, 987, 1597, are set of {9,16}_Φ_, Appendix A).

**Figure 3 micromachines-14-02152-f003:**
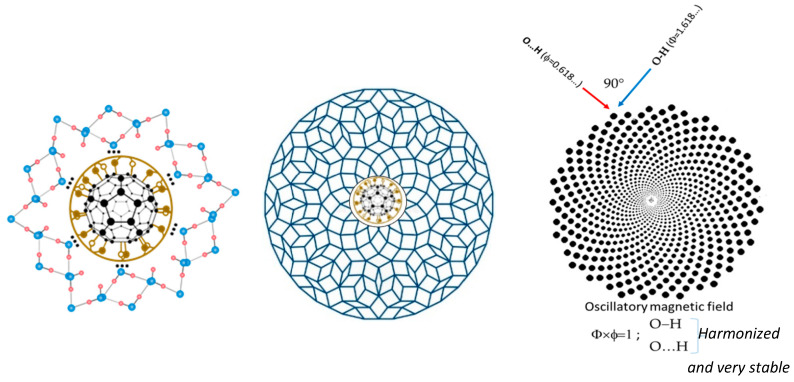
The first water layer of “icosahedar water base-tiles (only 6 are visible in cros section) interacting with FD-C_60_ (fullerol) (**left**). Order of 9 or 16 different “water icosahedral base-tiles” of Φ in the system of water layers (adapt from [4]) (**middle**) and the harmony of the oscillatory process of hydrogen bonds in 3DPT system as attractor of O-H and O…H hydrogen bonds (**right**), where Φ × ϕ = 1.

**Figure 4 micromachines-14-02152-f004:**
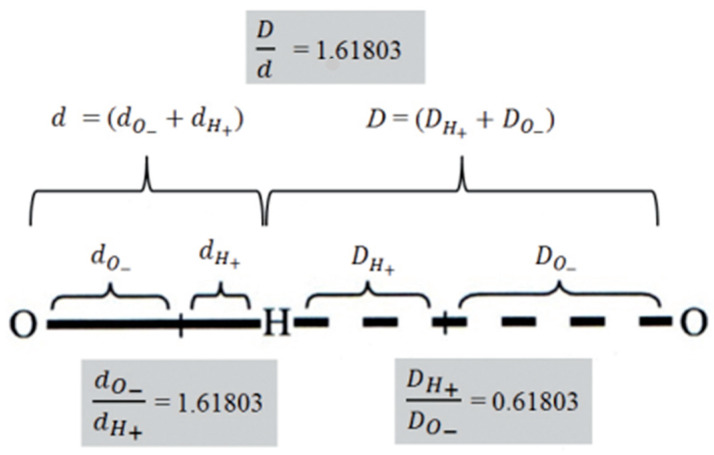
Nano machinery of charge distribution in the 3DPT system of covalent and non-covalent hydrogen bonds of water molecules. The hydrogen bonds are, actually, one of the best examples of the Yin–Yang concept, the concept of an ordered quadruple: big Yin (do_−_), small yin (d_H+_), big Yang (D_O−_), small yang (D_H+_). Adapted from [6].

**Figure 5 micromachines-14-02152-f005:**
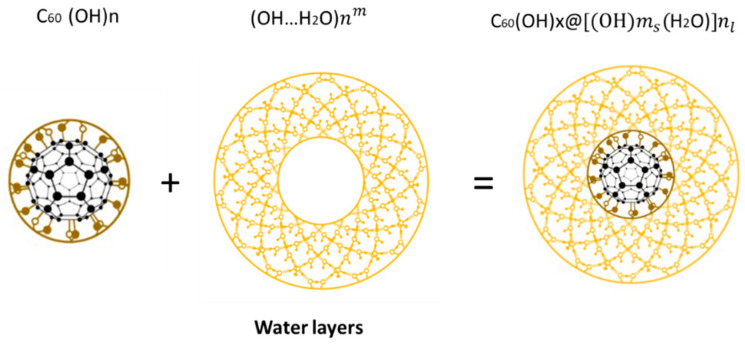
FD-C_60_ (fullerol, C_60_ (OH)_24–48_) is composed of molecule C_60_ and OH_24–48_ groups (average 36) (**left**), water layers as shells of 28, 60, 100, 280, 320 nm … surrounding FD-C_60_ (**middle**), and SD-C_60_ as soft-matter structure; size 3–30 nm depends on shell number (**right**).

**Figure 6 micromachines-14-02152-f006:**
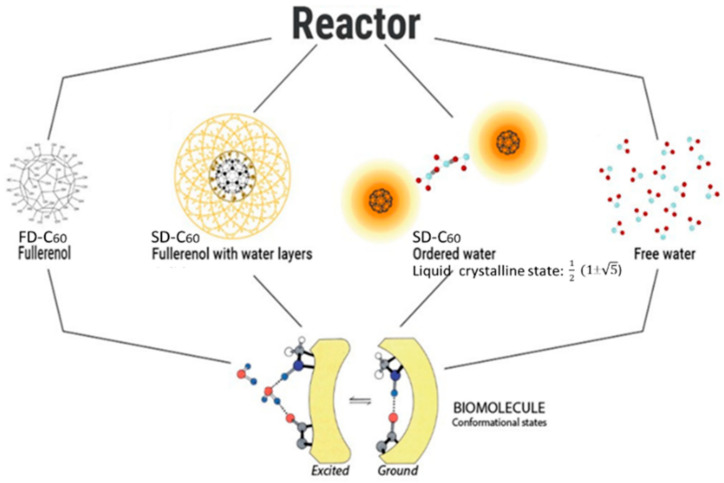
The input substances in the magnetic reactor are FD-C_60_ (fullerol or fullerenol) and highly pure water, while the output substances are SD-C_60_ in a solid state (2.5–3%); ordered water in chains between solid SD-C_60_ of about 58–60%; some small percentage (about 0.05%) of FD-C_60_ (around which water layers failed to form); and free water, 36–38%.

**Figure 7 micromachines-14-02152-f007:**
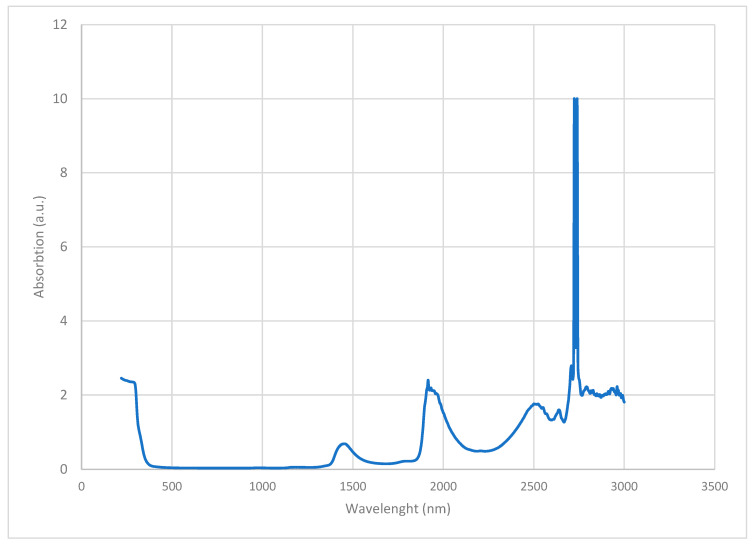
UV-Vis-NIR spectra of SD-C_60_-Y (3HFWC) in the 250–300 nm domain. It can be seen that there are two very close peaks of hydrogen bonds, at 2724 nm and 2740 nm. The peaks at 1800 nm and 1400 nm are the first and second overtones of the aqueous solution of SD-C_60_ and water. The region (1300–1500 nm), around peak 1400 nm, is important to determinate number of water layers (shells).

**Figure 8 micromachines-14-02152-f008:**
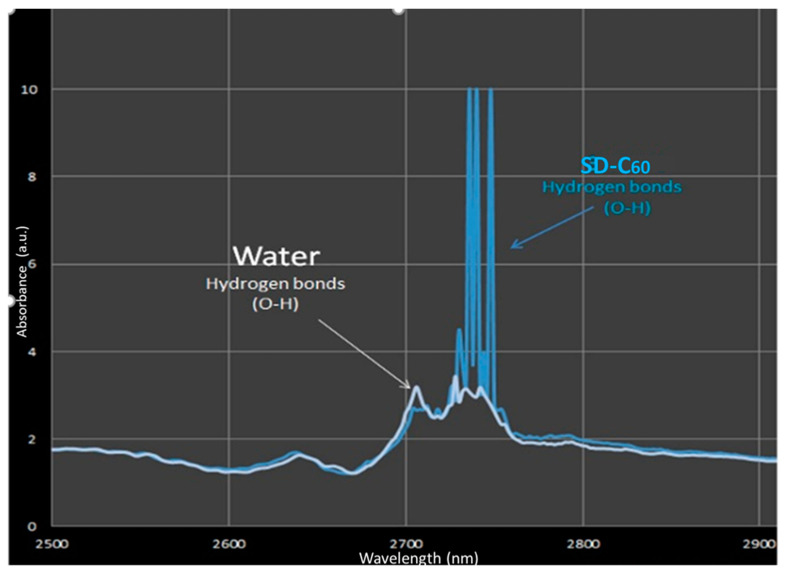
SD-C_60_-W (3HFWC-W) NIR spectra of SD-C_60_-Y (3HFWC-W) and water in the 2500–3000 nm range. It can be seen that there is a similarity between water and SD-C_60_ spectra in domains 2500–2720 nm and 2750–3000 nm, while in domain 2720–2750 nm, there are three close peaks of SD-C_60_ at 2721 nm, 2729 nm, and 2748 nm. Intensity of hydrogen bonds of SD-C_60_-W is three times higher than pure water hydrogen bonds.

**Figure 9 micromachines-14-02152-f009:**
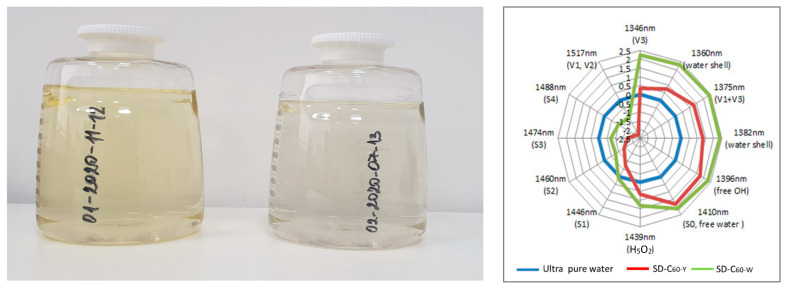
Two types of SD-C_60_: (1) SD-C_60_-Y (3HFWC: 01-2020-11-12) with light-yellow color from small presence of FD-C_60_ (fullerol) in solution and (2) SD-C_60_-W (3HFWC-W: 02-202007-13) white color without presence of FD-C_60_ (**left**). IR spectra (domain of the second water overtone in form of spider net) of SD-C_60_-Y and SD-C_60_-W in wavelength domain 1340–1517 nm, where 1360 nm and 1382 nm show different number of water layers (shells). Since maximal number of layers is 9 (what is case for SD-C_60_-W), from diagram (**right**), it is evident that SD-C_60_-Y has 3, 4, or 5 water layers.

**Figure 10 micromachines-14-02152-f010:**
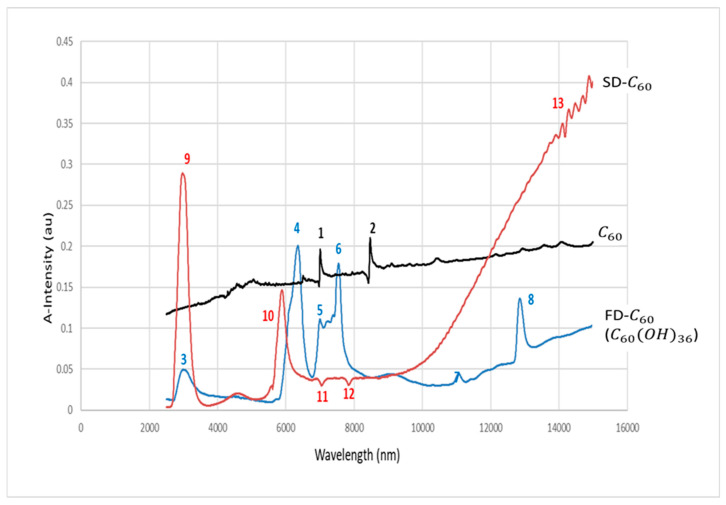
$FTIR\;Spectra\;of\;C_{60}$, FD-C_60_ $\left(Fullerol\;C_{60}{\left(OH\right)}_{36}\right)$ and SD-C_60_ ($C_{60}{\left(OH\right)}_{36}@{\left(H_{2}O\right)}_{1280}{\left(OH\right)}_{120}^{p}$) in domain 2500–14,000 nm. Peaks 1 and 2 are from C_60_ molecule, peaks 3–8 are from FD-C_60_, while peaks 9–13 are from SD-C_60_. Difference of intensity of hydrogen bonds of FD-C_60_ and SD-C_60_ is about 5.7 times (peaks 3 and 9).

**Figure 11 micromachines-14-02152-f011:**
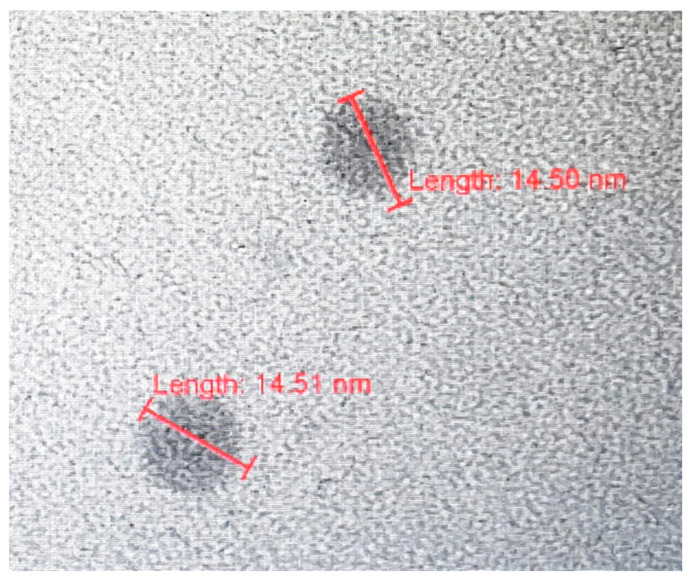
TEM images of dry SD-C_60_ (soft-matter), size about 15 nm. The solution of SD-C_60_ was three years old (stored at room temperature), which showed that soft-matter of SD-C_60_ is very stable in solution.

**Figure 12 micromachines-14-02152-f012:**
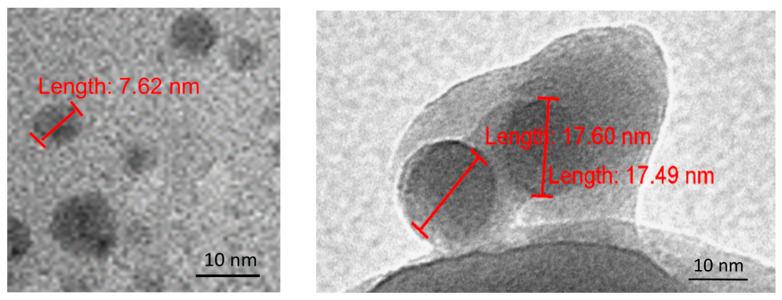
TEM images of dry SD-C_60_-Y (3HFWC, as a soft-matter), size 5–10 nm, 3 years old (before drying, it had stayed 3 years in solution) (**left**) and dry SD-C_60_-W (3HFWC-W, soft-matter), size 17–18 nm, old eight months (**right**).

**Figure 13 micromachines-14-02152-f013:**
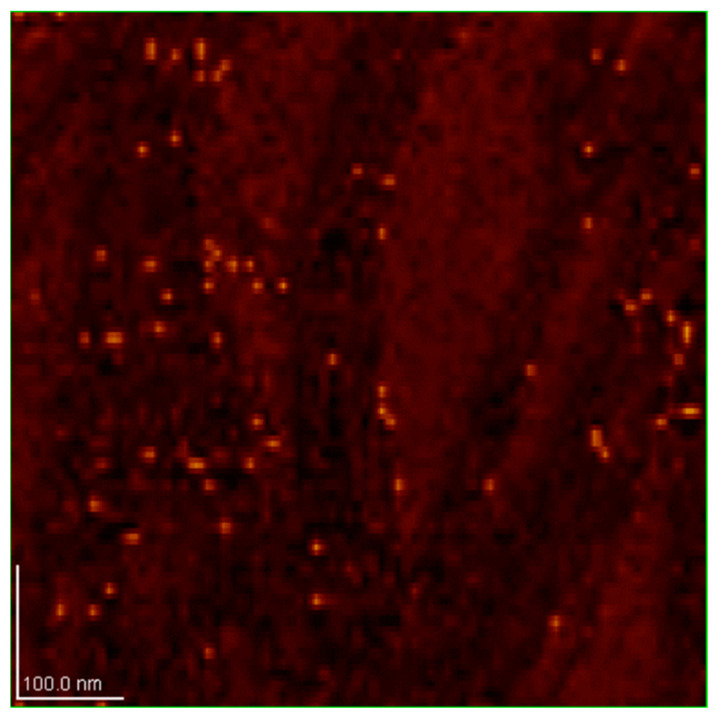
MFM image of dray SD-C_60_ (soft-matter), scan size 1.0 × 1.0 μm. On this image SD-C_60_ are well observed but without precise determination of size.

**Figure 14 micromachines-14-02152-f014:**
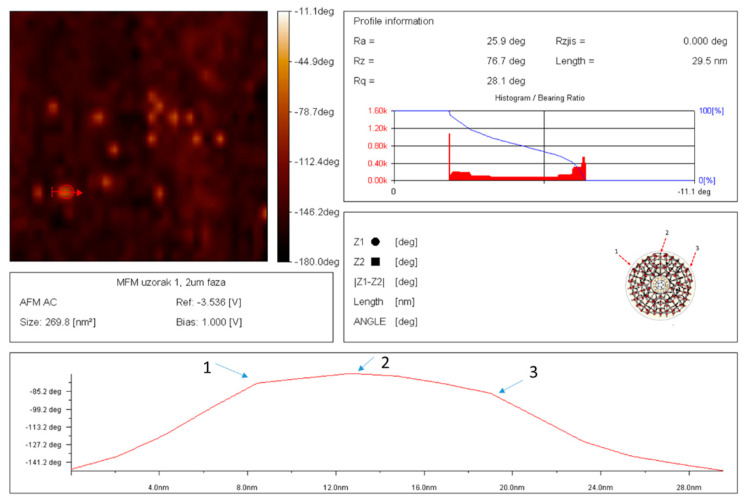
AFM/MFM image of dray SD-C_60_-W (3HFWC-W) with precise determination, size 29.5 nm, intensity of dipole–dipole interaction is determinate by angle of declination −80 deg.; this means that SD-C_60_ are rich with dipole interactions (water molecules). It can be seen from the picture that the line that determines the size of the SD-C_60_ as a soft-matter structure is not ideally smooth, but two sudden changes of direction (8 nm and 19 nm) can be observed. The reason for this is that the dry SD-C_60_ substance is not an ideal sphere but rather an icosahedral symmetry structure that has small “mild breaking points“, position 1, 2, and 3.

**Figure 15 micromachines-14-02152-f015:**
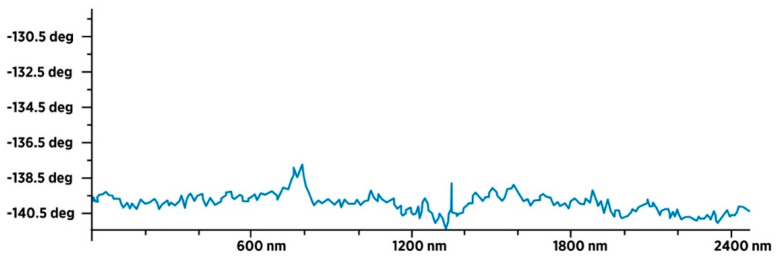
MFM spectra of dray FD-C_60_ (fullerol). Intensity of paramagnetic spectra is very small because presence of water is very low, only from humidity.

**Figure 16 micromachines-14-02152-f016:**
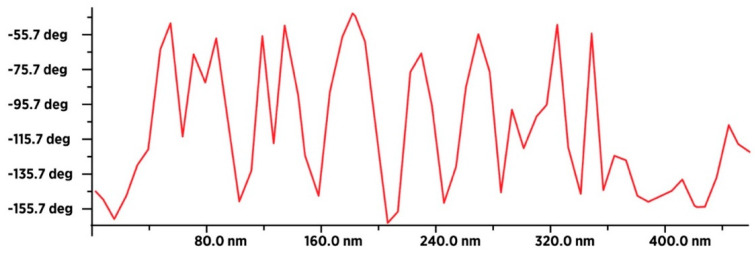
MFM spectra of dry SD-C_60_-W (3HFWC-W). Intensity of paramagnetic spectra is very high which means that 3HFWC-W is very rich in water molecules that interact with MFM tip (dipole–dipole interaction).

**Figure 17 micromachines-14-02152-f017:**
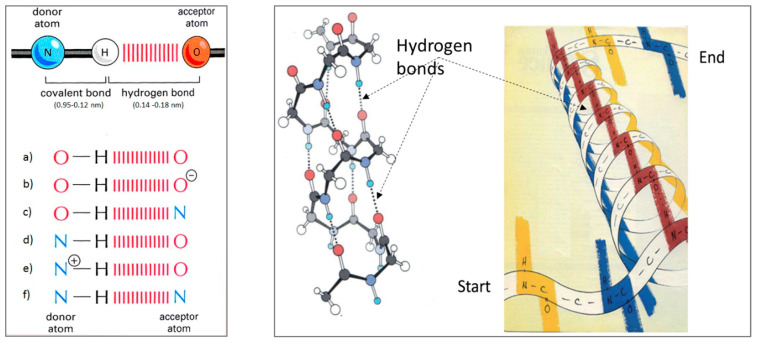
Six basic types of hydrogen bonds in biological systems (**left**) and an example of the arrangement of hydrogen bonds in the collagen structure (**right**). It can be seen that collagen (which has 100% an a-helix secondary structure) has three chains of hydrogen bonds. Any disturbance in the vibrations of hydrogen bonds leads to a greater or lesser degree of collagen dysfunction. (Adapted from [33]).

**Figure 18 micromachines-14-02152-f018:**
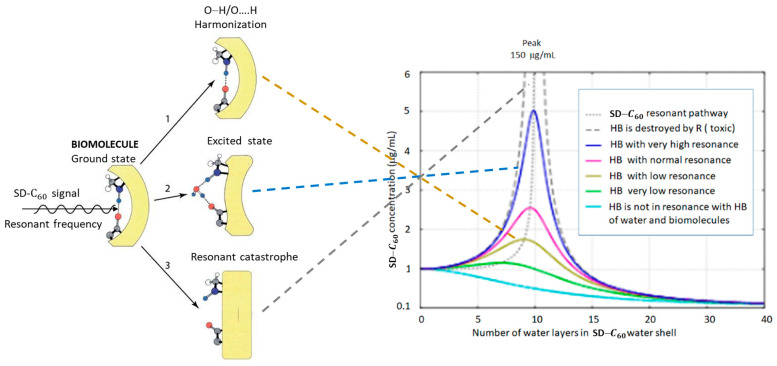
When the SD-C_60_ vibrations, transmitted through the hydrogen bonds of water, reach biomolecules (that have hydrogen bonds in their structure), then three basic cases can occur: (1) harmonization of the operation of the hydrogen bond (if the bond is broken, it will be re-established and lead the structure to a normal conformational state), (2) excitation of biomolecules by bringing water molecules to the place where the hydrogen bond was (there is a change in the conformational state of the biomolecule at that place), and (3) breaking of the hydrogen bond and dysfunction of the biomolecule. Intensity of action depends on SD-C_60_ concentration (in soft-matter stae) and number of water layers in icosahedral water onion. Better functionality of hydrogen bonds can be realized if they are acted upon not only with SD-C_60_, but when combined with gold nano particles (GNP) (Appendix A).

**Figure 19 micromachines-14-02152-f019:**
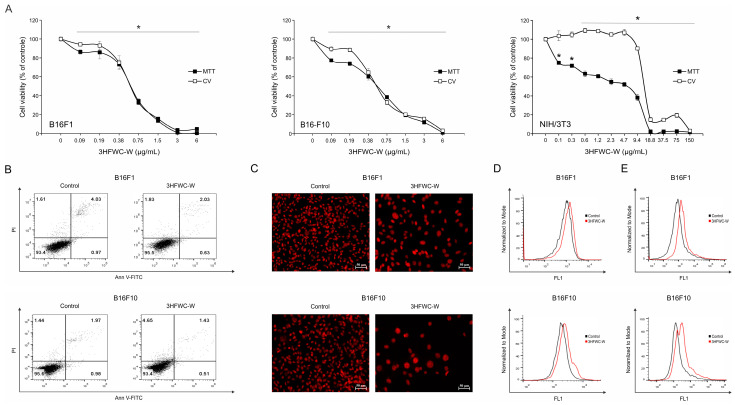
Antimelanoma action of 3HFWC-W (**A**) The viability of B16F1, B16F10, and NIH/3T3 cells treated with a wide range of concentrations of 3HFWC-W for 24 h was determined by MTT and CV assays. Cell viability is represented as a percentage of the absorbance value of the control culture grown in the medium, which was arbitrarily assigned a viability value of 100%. Results represent the mean ± SD of one representative out of three independently performed experiments. * *p* < 0.05 compared with control. (**B**) The percentage of early (Ann^+^/PI^−^) and late (Ann^+^/PI^+^) apoptotic cells, (**C**) morphology of cell nuclei, (**D**) caspase activity, and (**E**) proliferation rate in B16F1 and B16F10 melanoma cell cultures treated with the IC_50_ value of 3HFWC-W for 24 h were determined by Ann V-FITC/PI, PI, ApoStat, and CFSE staining, respectively, and subsequent flow cytometry (**B**,**D**,**E**) and fluorescence microscopy ((**C**), orig. magnification 200×). Representative dot plots, histograms, and micrographs of one out of three independently performed experiments are shown.

**Figure 20 micromachines-14-02152-f020:**
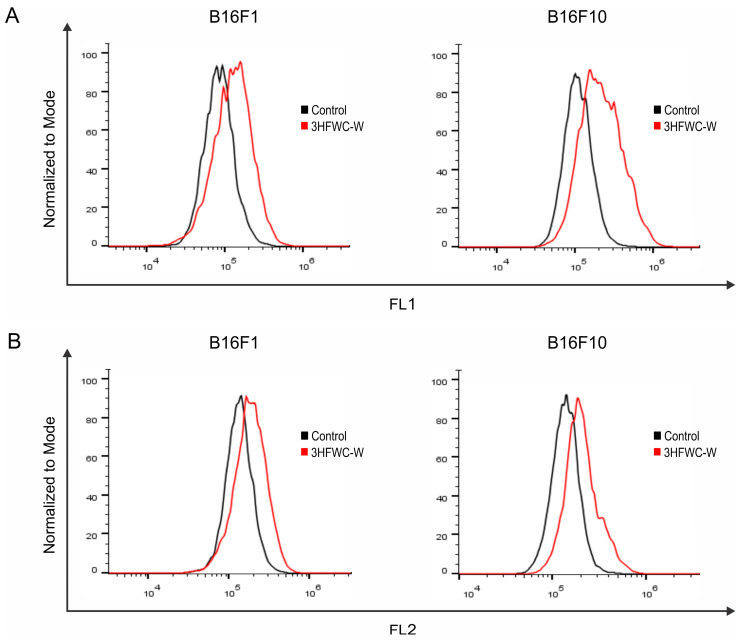
The effect of 3HFWC-W on intracellular ROS/RNS production in melanoma cells. Fluorescence intensity of B16F1 and B16F10 cells treated with IC_50_ value of 3HFWC-W for 24 h was determined by (**A**) DHR 123 and (**B**) DHE staining and subsequent flow cytometry. Representative histograms of one out of three independently performed experiments are shown.

**Figure 21 micromachines-14-02152-f021:**
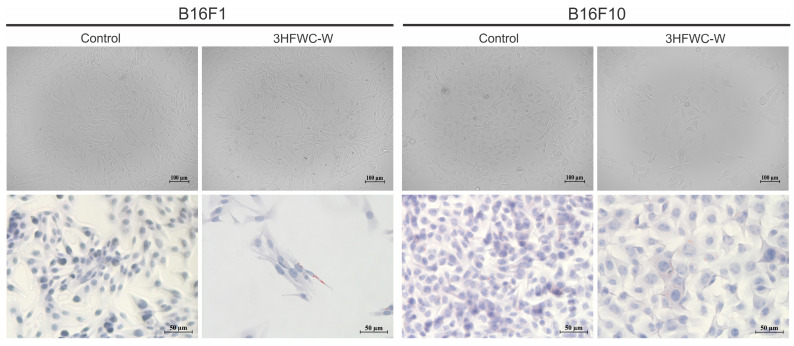
The effect of 3HFWC-W on the morphology and phenotype of melanoma cells. Morphological and intracellular changes of B16F1 and B16F10 cells treated with the IC_50_ value of 3HFWC-W for 24 h was determined without staining (upper panel micrographs) and by Oil Red O staining (lower panel micrographs, orig. magnification 200×) and subsequent light microscopy. Representative micrographs are shown.

**Table 1 micromachines-14-02152-t001:** IC_50_ values of 3HFWC-W in all tested cell lines obtained from MTT and CV assays ^1^.

Cell Line	Assay	${IC}_{50}\;(\mu g/\mathrm{mL})$ 3HFWC
B16F1	MTT CV	0.60 ± 0.00 0.68 ± 0.10
B16F10	MTT CV	0.65 ± 0.16 0.77 ± 0.29
NIH/3T3	MTT CV	5.40 ± 0.01 14.40 ± 0.50

^1^ Data are presented as the mean ± standard deviation of the mean (SD) of three independent experiments.

## Data Availability

Data are contained within the article.

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
