# Peer review of "The Second Derivative of Fullerene C60 (SD-C60) and Biomolecular Machinery of Hydrogen Bonds: Water-Based Nanomedicine"

_micromachines, 2023, doi:10.3390/mi14122152_

Round 1

Reviewer 1 Report

Comments and Suggestions for Authors

The authors of the presented manuscript examined the Fullerene C60 derivative. They used Fullerenol (FD-C60) - a derivative of Fullerene C60 obtained by its hydroxylation. It was shown that such a derivative possesses an average of 36 hydroxyl groups and has decreased cytotoxic effects in comparison to C60, but FD-C60 is still toxic for living cells. Therefore, the authors created the second derivative of the C60 molecule (SD-C60) containing water layers around the FD-C60. Then, they used various methods to elucidate the structure of the obtained nanoparticles. In my opinion, these results are convincing and allow for a good description of analyzed nanoparticles, especially water layers.

Then, the authors tested the biological activity of analyzed two types of nanoparticles against melanoma cells. They showed that the mechanism of action of these nanoparticles is similar, and they hypothesize that induction of cell senescence may play a main role in this mechanism.

The evaluated manuscript is very extensive. In my opinion, it would be divided into two parts - physical-chemical analyses of manufactured nanoparticles and examination of their biological action. However, it is the editor's decision, which capacity of the article may be acceptable. The authors described the toxicity of pure Fullerene C60 and its derivatives. However, they skipped an aqueous solution of Fullerene C60, which was described as non-toxic against various cells. Moreover, one of the important manners of biological activity of such Fullerene C60 form is water layers that surround nanoparticles. In my opinion, such a form of Fullerene C 60 should be also discussed in the evaluated manuscript.

Author Response

Dear Sir/Madam,
Thank you for the review of the work and the comments that needed to be clarified. We have done it and we hope that we have resolved the dilemmas. Attached is the answer to your questions . The paper with supplementary material where the changes have been included is sent to the Editor (because it could not upload.
Regards.
The Authors

Reviewer 2 Report

Comments and Suggestions for Authors

The manuscript presents a report on the synthesis and characterization of two forms of SD-C60 with different number of water shells (3HFWC and 3HFWC-W). The existence of water shells exhibited significant cytotoxic activity and improved effects on biological water and biomolecules. More interestingly, the improved formulation 3HFWC-W will suppress cell growth mainly via the induction of senescence, which is a more desirable mechanism of antitumor action relative to cell death, thus exhibiting great potential in advanced medical treatment. On this basis, the author introduces an effective protocol for the treatment of various pathologies by the use of nano SD-C60 substance as “water-based nanomedicine”. Considering these significant findings, I believe the manuscript has the potential to be published, provided the following concern is addressed.

1.     My major concern is that the author claims that 3HFWC-W (6-9 water layers) realized anti-melanoma action basically through the initiation of cell reprogramming, whereas 3HFWC (1-5 water layers) suppresses cell growth mainly attribute to the contribution of cell death. Why does the increase of water layers for two SD-C60 derivatives change the mode of action? What is the role of water layers in altering the mechanism from cell death to induction of senescence?

2.     In figure 12, the length of dray 3HFWC is measured as 7.62 nm, and the length of dray 3HFWC-W is measured as 17.60 nm (more than twice of 3HFWC). However, it is obviously to see that the length of dray 3HFWC in left image is even longer than the half of the length of 3HFWC-W in right image. Please double check the length of derivatives.

Author Response

Dear Sir/Madam,
Thank you for the review of the work and the comments that needed to be clarified. We have done it and we hope that we have resolved the dilemmas. Attached is the answer to your questions. The paper with the supplementary material where the changes were included was sent to the Editor because they could not be approved by you.
Regards.
The Authors

Round 2

Reviewer 1 Report

Comments and Suggestions for Authors

In my opinion, the article should be accepted for publication in its current form.